# CONTINUOUS TEST-TIME ADAPTATION OF VISION-LANGUAGE MODELS

## ABSTRACT

Test-time adaptation (TTA) has emerged as a promising paradigm for bridging the distribution gap between pretraining and test data in vision language models (VLMs). Unfortunately, existing methods either assume a static target-domain distribution or rely only on a small subset of samples, which fails to adapt the continuous real-world distributions. In this work, we propose Continuous Test-Time Adaptation (C-TTA), which adapts to the entire target-domain distribution via a continuously updated target prototype that adaptively incorporates visual features from incoming unlabeled test samples based on their class confidence. It is worth highlighting that C-TTA updates only a simple target prototype, which circumvent the heavy backpropagation and large cache access required by previous methods. This endows C-TTA with extremely high efficiency while achieving state-of-the-art performance on 15 image classification benchmarks. For example, C-TTA outperforms all existing training-required methods in cross-dataset generalization, while achieving $5.7\times$ faster inference than cache-based TDA on ImageNet. Beyond image classification, C-TTA can be easily applied to 3D VLMs, achieving significant performance gains on 4 challenging point cloud analysis benchmarks.

## 1 INTRODUCTION

The rapid progress of pre-trained vision language models (VLMs) has brought substantial advances to the computer vision community (Jia et al., 2021; Radford et al., 2021a; Alayrac et al., 2022; Xue et al., 2023; 2024). However, their deployment in real-world applications is often hindered by inherent distribution shifts between pre-training data and test data. To address this challenge, prior efforts have typically leveraged few-shot downstream data to adapt VLMs to downstream tasks, either by tuning prompts (Zhou et al., 2022b; Chen et al., 2023; Fu et al., 2024) or by using visual adapters (Gao et al., 2024; Zhang et al., 2022b; Udandarao et al., 2023). Nevertheless, such approaches remain limited by their reliance on high-quality, task-specific annotations, which are rarely feasible in real-world applications.

In response, recent works (Zhang et al., 2022a; Wang et al., 2021) have investigated test-time adaptation (TTA), which enables VLMs to align with unlabeled test samples, thereby mitigating distribution shifts and reducing dependence on labeled data. Existing approaches can be broadly divided into two categories: (1) **Instance-wise TTA** (Figure 1(I)(a)), which adapts each test instance independently by optimizing text prompts (Shu et al., 2022; Feng et al., 2023; Zhang et al., 2024b; Abdul Samadh et al., 2023) or distributional biases (Huang et al., 2025) with unsupervised objectives (e.g., entropy minimization). Although effective in alleviating distribution gaps, this paradigm rests on the unrealistic assumption of static target distributions, which fails in dynamic real-world scenarios. (2) **Episodic-wise TTA** (Figure 1(I)(b)), which preserves high-confidence sample information in caches (Karmanov et al., 2024; Zhang et al., 2024a) or priors (Zhou et al., 2025), and access them on-the-fly during inference to improve model predictions. However, such methods presuppose access to only a small subset of test data, which remains insufficient for coping with continuously evolving real-world distributions.

To sum up, existing approaches either overlook the dynamic nature of the real-world or rely solely on high-confidence samples, both of which fail to faithfully model continuous target distributions. This paper introduces **Continuous Test-Time Adaptation (C-TTA)**, a novel TTA method that enables VLMs to continuously leverage all test samples in dynamic environments to capture the target-

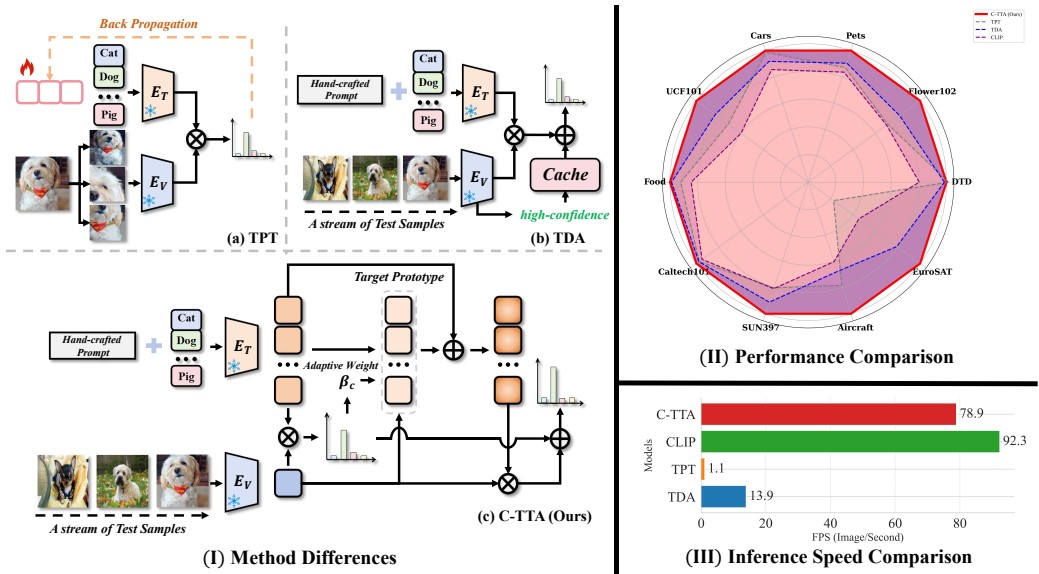

Figure 1: **Overview and effectiveness of C-TTA.** (I) Method comparison. (a) Instance-wise TTA (e.g., TPT (Shu et al., 2022)) and (b) Episodic-wise TTA (e.g., TDA (Karmanov et al., 2024)) fail to capture the continuous target-domain distribution. (c) In contrast, our proposed C-TTA introduces a simple target prototype that continuously aggregates information from all test samples, enabling faithful modeling of the entire target domain. (II) Performance comparison on 10 unseen cross-dataset benchmarks, where C-TTA consistently outperforms prior approaches. (III) Inference speed comparison on ImageNet, showing that C-TTA achieves nearly the same efficiency as CLIP (Radford et al., 2021a) while being 5.7× faster than TDA.

domain distribution. As illustrated in Figure 1(I)(c), the core of C-TTA is a target prototype that continuously collects knowledge from online unlabeled samples. Specifically, we adaptively incorporate the visual feature of each test sample into the target prototype based on its class confidence, and then fuse it with the pretrained text prototypes. This design allows the model to preserve pretrained knowledge while continuously adapting to the evolving target distribution, which translates into leading performance for C-TTA, as illustrated in Figure 1(II). Beyond performance, C-TTA is also more efficient, since it operates without backpropagation and also avoids the heavy storage and access of test samples. For example, as shown in Figure 1(III), it achieves 5.7× faster inference than TDA and 71.7× faster than TPT on ImageNet.

To summarize our contributions: (1) We propose Continuous Test-Time Adaptation (C-TTA), which dynamically models the continuous target distribution via a simple target prototype. (2) C-TTA is training-free and cache-free, significantly accelerating existing TTA methods. (3) Extensive experiments demonstrate the effectiveness of C-TTA in improving the zero-shot performance of VLMs.

## 2 RELATED WORKS

### 2.1 CONTRASTIVE VISION-LANGUAGE MODELS

Contrastive vision-language models (Radford et al., 2021a; Jia et al., 2021; Xue et al., 2023; 2024) achieve effective zero-shot visual understanding by learning rich cross-modal correlations from large-scale visual-text pairs. However, the inherent distribution gap between pretraining data and downstream test environments may cause substantial performance degradation. Existing methods Zhou et al. (2022a); Khattak et al. (2023); Zhang et al. (2022b); Gao et al. (2024) typically address this issue by collecting a small amount of labeled downstream data to fine-tune VLMs for task adaptation. For instance, CoOp Zhou et al. (2022b) replaces handcrafted prompts with learnable context tokens for task adaptation, while TextRefiner Xie et al. (2025) further incorporates internal

image knowledge into optimized vectors. While effective, these methods require labeled data and incur extra training costs, limiting real-world applicability. In contrast, in this work, we investigate test-time adaptation for VLMs, which enables models to adapt to downstream data without requiring any labeled data.

## 2.2 TEST-TIME ADAPTATION FOR VLMS

Test-time adaptation (TTA) aims to align pretrained models with new data distributions at test time, without requiring additional labeled data. Recently, this practical paradigm has been explored in VLMs (Zhang et al., 2024a; Zanella & Ben Ayed, 2024; Huang et al., 2025). As a pioneering work, TPT (Shu et al., 2022) adjusts learnable prompts by enforcing consistency across different augmented views of each test sample, while DiffTPT (Feng et al., 2023) further enhances generalization by introducing a richer augmentation via diffusion models. However, these approaches treat each test sample independently, effectively resetting the model for every new instance. This implicitly assumes a static target distribution, which rarely holds in dynamic real-world scenarios. An alternative line of work treats test data as an online stream. For example, TDA (Karmanov et al., 2024) constructs positive and negative caches to store high-confidence samples and high-uncertainty samples, respectively, and leverages cache retrieval to enhance model predictions during testing. Unlike TDA, BCA Zhou et al. (2025) initializes an empty class prior and incrementally updates it using the posterior information from incoming high-confidence samples. While effective, these methods consider only a small subset of test samples, which remains insufficient for modeling the continuity of real-world target distributions. In this work, we propose Continuous Test-Time Adaptation (C-TTA), which introduces a target prototype to model the entire target-domain distribution by continuously aggregating visual information from all test samples. Notably, our approach eliminates the need for backpropagation or cache access, achieving significant improvements in generalization while maintaining high inference efficiency.

## 3 METHOD

### 3.1 BACKGROUND

**Contrastive Vision Language Models.** By jointly embedding images and text into a shared latent space, contrastive vision language models (Radford et al., 2021a; Jia et al., 2021; Alayrac et al., 2022) achieve rich cross-modal representations. Without loss of generality, we adopt CLIP (Radford et al., 2021b) as the foundational model for our methodological demonstration. CLIP is composed of a visual encoder $\mathbf{E_V}$ and a text encoder $\mathbf{E_T}$, which extract features from images and text, respectively. During training, a contrastive loss (Chen et al., 2020) is used to maximize similarity between matched image-text pairs while minimizing it for non-matching pairs.

At test time, given an input image $\boldsymbol{x}$, the visual encoder produces a visual feature $\boldsymbol{F}_g = \mathbf{E_V}(\boldsymbol{x}) \in \mathbb{R}^d$, where $d$ is the feature dimension. For a specific downstream task with $C$ classes, each class is incorporated into a hand-crafted prompt, yielding the corresponding text embeddings $\{\boldsymbol{F}_t^c\}_{c=1}^C$, where $\boldsymbol{F}_t^c \in \mathbb{R}^d$ denotes the text embedding of the class-specific text input. The prediction probability of image $\boldsymbol{x}$ belonging to class $y_c$ is then computed from the cosine similarity between the visual feature and the text embeddings, expressed as:

$$p_{\text{CLIP}}(y_c|\boldsymbol{x}) = \frac{\exp\left(\cos\left(\boldsymbol{F}_g, \boldsymbol{F}_t^c\right)/\tau\right)}{\sum_{c=1}^C \exp\left(\cos\left(\boldsymbol{F}_g, \boldsymbol{F}_t^c\right)/\tau\right)}, \tag{1}$$

where $cos(\cdot)$ calculates the cosine similarity between vectors, and $\tau$ controls the sharpness of the softmax distribution.

**Revisiting Test-time Adaption of VLMs.** The output of CLIP (as shown in Eq. 1) supports effective zero-shot image classification. However, its performance can be severely limited when test data undergo distribution shifts from the training domain. In response this drawback, recent research has introduced test-time adaptation (Shu et al., 2022; Feng et al., 2023; Karmanov et al., 2024; Zhou et al., 2025), aiming to mitigate distribution shifts and enhance model robustness. To achieve this, one common paradigm is to use multiple augmented views of the test image, denoted as $\{\mathcal{A}_n(\boldsymbol{x})\}_{n=1}^N$, where $\mathcal{A}(\cdot)$ is the augmentation function, and $N$ denotes the number of augmented

views. As a prominent approach, TPT (Shu et al., 2022) adapts text prompts by enforcing prediction consistency across different augmented views. Particularly, the augmented views are firstly leveraged to produce a set of confident logits, formulated as:

$$\tilde{p}\left(y_c|\boldsymbol{x}\right) = \frac{1}{\rho N} \sum_{i=1}^{N} \mathbb{I}\left[\mathbf{H}\left(p_i\right) \leq \theta\right]\left(p_i\left(y_c \mid \mathcal{A}_i(\boldsymbol{x})\right)\right), \tag{2}$$

where $\mathbf{H}(\cdot)$ denotes the self-entropy of the predicted probability distribution, and the indicator function $\mathbb{I}\left[\mathbf{H}\left(p_i\right) \leq \theta\right]$ retains only logits with uncertainty below the threshold $\theta$. Afterwards, TPT optimizes the text prompt $\boldsymbol{V}$ by minimizing the entropy of the distribution over the top-$\rho$ most confident samples $\tilde{p}\left(y_c|\boldsymbol{x}\right)$ as:

$$\mathbf{H}\left(\tilde{p}\right) = -\sum_{c=1}^{C} \tilde{p}_g\left(y_c \mid \boldsymbol{x}\right) \log \tilde{p}_g\left(y_c \mid \boldsymbol{x}\right), \tag{3}$$

$$\boldsymbol{V}^* \leftarrow \boldsymbol{V} - \eta \nabla_{\boldsymbol{V}} \mathbf{H}\left(\tilde{p}\right), \tag{4}$$

where $\eta$ represents the learning rate for gradient descent, while $\nabla_{\boldsymbol{V}} \mathbf{H}\left(\tilde{p}\right)$ denotes the gradient of the entropy. The optimized prompt $\boldsymbol{V}^*$ serves as CLIP's new text input, generating the corresponding test-time text embeddings $\{\boldsymbol{F}_{\boldsymbol{V}^*}^c\}_{c=1}^C$. Consequently, the output of TPT can be expressed as:

$$p_{\text{TPT}}(y_c|\boldsymbol{x}) = \frac{\exp\left(\cos\left(\boldsymbol{F}_g, \boldsymbol{F}_{\boldsymbol{V}^*}^c\right)/\tau\right)}{\sum_{c=1}^{C} \exp\left(\cos\left(\boldsymbol{F}_g, \boldsymbol{F}_{\boldsymbol{V}^*}^c\right)/\tau\right)}. \tag{5}$$

Despite this paradigm can improve test-time performance, these **instance-wise TTA** methods typically reset the model after adapting to each sample, rendering it blind to previously seen test data. This assumes a static target distribution, which fails to reflect the dynamic nature of real-world scenarios. Subsequent methods (Karmanov et al., 2024; Zhang et al., 2024a) treat test data as an online stream and propose to improve predictions using a subset of high-confidence samples, referred to as **episodic-wise TTA**. A widely practice is to store these high-confidence samples in a cache $\boldsymbol{Q}_{cache} \in \mathbb{R}^{KC \times d}$, where $K$ denotes the max cache size allocated for each class. At test time, test sample $\boldsymbol{x}$ serves as a query to retrieve similar entries from the cache as auxiliary information, which can be expressed as:

$$p_{\text{cache}}(\boldsymbol{x}) = A\left(\boldsymbol{F}_g Q_{\text{cache}}^T\right) L_{cache}, \tag{6}$$

where $A(z) = \alpha \exp(-\beta(1-z))$ is a scaling function within a weighting factor $\alpha$ and a sharpness ratio $\beta$. $L_{cache} \in \mathbb{R}^{KC \times C}$ denotes the one-hot vectors of the pseudo-label predicted by CLIP. The final prediction is obtained by integrating the CLIP prediction $p_{\text{CLIP}}$ with the cache-based prediction $p_{\text{cache}}$. In contrast to the instance-wise TTA, these approaches provide a dynamic view of the target domain by leveraging previously seen samples. Unfortunately, they consider only a very small set of test data, limiting test-time adaptation's ability to fully utilize continuously incoming test samples.

### 3.2 CONTINUOUS TEST-TIME ADAPTATION

In this work, we introduce a continuous test-time adaptation (C-TTA) designed to handle the continuously evolving distributions of real-world target domains. The core contribution of C-TTA lies in introducing a continuously updated target prototype. As illustrated in Figure 1(I)(c), this prototype adaptively integrates the visual features of incoming test samples according to their class confidence, thereby accumulating knowledge of the target domain over time. Notably, our approach eliminates the need for explicit sample storage or retrieval, while also avoiding reliance on backpropagation. We describe the detailed implementation of C-TTA below.

We begin by zero-initializing the target prototype $\mathbf{P}_t \in \mathbb{R}^{C \times d}$, whose dimensionality is aligned with that of the text prototypes $\mathbf{F}_t$ obtained from CLIP. It is further represented as a set of class-specific prototypes:

$$\mathbf{P}_t = \left\{\mathbf{P}_t^1, \mathbf{P}_t^2, \cdots, \mathbf{P}_t^C\right\}. \tag{7}$$

At test time, given an incoming sample $\boldsymbol{x}$, we can obtain its visual feature $\boldsymbol{F}_g$ and the prediction confidence $p_{\text{CLIP}}(y_c|\boldsymbol{x})$ over all $C$ classes according to Eq. 1. For convenience, we collect these confidences into a single vector $\boldsymbol{s} \in \mathbb{R}^{1 \times C}$, where each entry $\boldsymbol{s}_j = p_{\text{CLIP}}(y_j|\boldsymbol{x})$. A natural intuition is that if a test sample exhibits a high confidence toward a particular class, it should contribute more strongly to the corresponding class-specific prototype, since such a sample is more likely to provide

reliable knowledge about the target domain. To realize this, we introduce an adaptive fusion weight $\beta_j$ for each class $j$, which is computed as follows:

$$\beta_j = 1 - e^{-\boldsymbol{s}_j/h}, \tag{8}$$

where $h$ is a temperature parameter that controls the smoothness of the weighting. Subsequently, we inject the visual feature $\boldsymbol{F}_g$ of the test sample into the target prototypes $\mathbf{P}_t$ by leveraging the adaptive fusion weights $\beta_j$, where each prototype $\mathbf{P}_t^j$ is updated in proportion to its corresponding confidence. The update rule can thus be formulated as:

$$\mathbf{P}_t^j \leftarrow (1 - \beta_j)\,\mathbf{P}_t^j + \beta_j \boldsymbol{F}_g. \tag{9}$$

Here, $\leftarrow$ denotes updating the value of a variable. As successive test samples are incorporated, the target prototypes $\mathbf{P}_t$ continuously accumulate target-domain knowledge from the incoming data. To harmonize the integration of test-time information with CLIP's semantic space, we introduce a refined target prototype $\tilde{\mathbf{P}}_t$ initialized from the original text prototypes $\boldsymbol{F}_t$. By performing a simple interpolation, we ensure that the target-domain knowledge remains aligned with the pretrained semantic space:

$$\tilde{\mathbf{P}}_t = (1 - w)\,\mathbf{P}_t + w\tilde{\mathbf{P}}_t, \tag{10}$$

where $w$ is the coefficient hyperparameter. Analogous to Eq. 1, we first compute the target-domain class probability distribution $p_{\texttt{Target}}$ based on the refined target prototypes $\tilde{\mathbf{P}}_t$:

$$p_{\texttt{Target}}(y_c|\boldsymbol{x}) = \frac{\exp\left(\cos\left(\boldsymbol{F}_g, \tilde{\boldsymbol{P}}_t^c\right)/\tau\right)}{\sum_{c=1}^{C} \exp\left(\cos\left(\boldsymbol{F}_g, \tilde{\boldsymbol{P}}_t^c\right)/\tau\right)}. \tag{11}$$

This distribution is then combined with the original CLIP output distribution to yield the final C-TTA predictions, as formulated below:

$$p_{\texttt{C-TTA}}(y|\boldsymbol{x}) = p_{\texttt{CLIP}}(y|\boldsymbol{x}) + p_{\texttt{Target}}(y|\boldsymbol{x}). \tag{12}$$

In summary, C-TTA preserves pretrained distribution while continuously adapting to the evolving target distribution. Furthermore, its design is versatile and can be readily extended to other VLMs, including 3D VLMs such as ULIP (Xue et al., 2023) for point cloud recognition.

## 4 EXPERIMENTS

### 4.1 EXPERIMENTAL SETUP

**Datasets.** To comprehensively evaluate the effectiveness of our method, we conduct experiments on 15 image classification datasets and 2 point cloud recognition datasets.

For image classification, we adopt two widely used TTA benchmarks: Cross-Dataset Generalization and Domain Generalization. Under the Cross-Dataset Generalization benchmark, we evaluate performance on 10 diverse datasets spanning a wide spectrum of recognition tasks. These datasets include Flowers102 (Nilsback & Zisserman, 2008), OxfordPets (Parkhi et al., 2012), StanfordCars (Krause et al., 2013), FGVC-Aircraft (Maji et al., 2013), Food101 (Bossard et al., 2014), EuroSAT (Helber et al., 2019), UCF101 (Soomro, 2012), DTD (Cimpoi et al., 2014), SUN397 (Sun et al., 2020), and Caltech101 (Fei-Fei et al., 2004). For the Domain Generalization benchmark, we further test our approach on 4 challenging out-of-distribution (OOD) variants of ImageNet (Deng et al., 2009): ImageNetV2 (Recht et al., 2019), ImageNet-Sketch (Wang et al., 2019), ImageNet-A (Hendrycks et al., 2021b), and ImageNet-R (Hendrycks et al., 2021a).

Beyond 2D images, we also extend our evaluation to point cloud recognition with 3D VLMs. Specifically, we consider 4 widely used corrupted point cloud benchmarks: ModelNet-C (Ren et al., 2022) and three corrupted variants of ScanObjectNN-C (Uy et al., 2019). ModelNet-C introduces seven atomic corruption types, including adding global outliers, adding local outliers, removing global structures, removing local parts, rotation, scaling, and jittering. Following the same protocol, we apply these atomic corruptions to the 3 variants of ScanObjectNN to construct their corrupted counterparts (Sun et al., 2025).

Table 1: Comparison of C-TTA in Cross-Datasets Generalization using ViT-B/16 Dosovitskiy (2020) as the backbone. Bold values indicate the best performance and † denotes methods that require additional trainable parameters.

| Method | Venue | Flowers102 | DTD | Pets | Cars | UCF101 | Caltech101 | Food101 | SUN397 | Aircraft | EuroSAT | Average |
|---|---|---|---|---|---|---|---|---|---|---|---|---|
| CLIP-ViT-B/16 | ICML2021 | 67.28 | 44.44 | 88.06 | 65.28 | 65.03 | 92.94 | 83.82 | 62.59 | 23.82 | 41.38 | 63.46 |
| *Training-time Adaptation* | | | | | | | | | | | | |
| CoOp† | IJCV2022 | 68.71 | 41.92 | 89.14 | 64.51 | 66.55 | 93.70 | 85.30 | 64.15 | 18.47 | 46.39 | 63.88 |
| CoCoOp† | CVPR2022 | 70.85 | 45.45 | 90.46 | 64.90 | 68.44 | 93.79 | 83.97 | 66.89 | 22.29 | 39.23 | 64.63 |
| MaPLe† | CVPR2023 | 72.23 | 46.49 | 90.49 | 65.57 | 68.69 | 93.53 | 86.20 | 67.01 | 24.74 | 48.06 | 66.30 |
| *Instance-wise Test-time Adaptation* | | | | | | | | | | | | |
| TPT† | NIPS2022 | 68.98 | 47.75 | 87.79 | 68.50 | 68.04 | 94.16 | 84.67 | 65.50 | 24.78 | 42.44 | 65.10 |
| DiffTPT† | ICCV2023 | 70.10 | 47.00 | 88.22 | 67.01 | 62.67 | 92.49 | 87.23 | 65.74 | 25.60 | 43.13 | 65.47 |
| PromptAlign† | NIPS2024 | 72.39 | 47.24 | 90.76 | 68.50 | 69.47 | 94.01 | 86.65 | 68.50 | 24.80 | 47.86 | 66.92 |
| ZERO | NIPS2024 | 67.07 | 45.80 | 86.74 | 67.54 | 67.64 | 93.51 | 84.36 | 64.49 | 24.40 | 39.60 | 64.66 |
| MTA | CVPR2024 | 68.26 | 45.59 | 88.22 | 68.05 | 68.11 | 94.13 | 84.95 | 64.98 | 25.32 | 38.71 | 64.63 |
| GS-Bias† | ICML2025 | 71.94 | 46.10 | 90.38 | 67.33 | 67.59 | 94.60 | 86.09 | 67.40 | 26.49 | 52.42 | 67.03 |
| TCA | ICCV2025 | 73.33 | 46.16 | 89.53 | 65.33 | 72.38 | 93.69 | 85.31 | 65.92 | 24.87 | **70.43** | 68.69 |
| *Episodic-wise Test-time Adaptation* | | | | | | | | | | | | |
| TDA | CVPR2024 | 71.42 | 47.40 | 88.63 | 67.28 | 70.66 | 94.24 | 86.14 | 67.62 | 23.91 | 58.00 | 67.53 |
| DPE† | NIPS2024 | 75.07 | **54.20** | **91.14** | 67.31 | 70.44 | 94.81 | 86.17 | **70.07** | **28.95** | 55.79 | 69.40 |
| BoostAdapter | NIPS2024 | 71.66 | 45.69 | 89.51 | **69.30** | 71.93 | 94.77 | 87.17 | 68.09 | 27.45 | 61.22 | 68.68 |
| HisTPT† | NIPS2024 | 71.20 | 48.90 | 89.10 | 69.20 | 70.10 | 94.50 | **89.30** | 67.20 | 26.90 | 49.70 | 67.60 |
| BCA | CVPR2025 | 73.12 | 53.49 | 90.43 | 66.86 | 67.59 | 94.69 | 85.97 | 68.41 | 28.59 | 56.63 | 68.59 |
| C-TTA | Ours | **75.11** | 47.58 | 90.90 | 68.74 | **74.00** | **94.85** | 86.39 | 69.20 | 26.34 | 61.49 | **69.46** |

**Baselines.** We compare our proposed method, C-TTA, with several state-of-the-art approaches. For 2D VLMs, the comparison includes zero-shot CLIP (Radford et al., 2021a), 3 representative training-time adaptation methods (CoOp (Zhou et al., 2022b), CoCoOp (Zhou et al., 2022a) and MaPLe (Khattak et al., 2023)), and 12 test-time adaptation methods, including 7 instance-wise TTA (TPT (Shu et al., 2022), DiffTPT (Feng et al., 2023), PromptAlign (Abdul Samadh et al., 2023), ZERO (Farina et al., 2024), MTA (Zanella & Ben Ayed, 2024), GS-Bias (Huang et al., 2025), and TCA (Wang et al., 2025)) and 5 episodic-wise TTA (TDA (Karmanov et al., 2024), BoostAdapter (Zhang et al., 2024c), HisTPT (Zhang et al., 2024b), DPE (Zhang et al., 2024a), and BCA (Zhou et al., 2025)), all tailored for vision-language models. Notably, CoOp, CoCoOp and MaPLe are trained on the ImageNet (Deng et al., 2009) dataset, with 16-shot per class, and then transferred to downstream datasets for evaluation. In contrast, test-time adaptation methods do not require access to the training data. For 3D VLMs, our evaluation includes comparisons with ULIP (Xue et al., 2023), a representative zero-shot method for point cloud recognition, and Point-Cache (Sun et al., 2025), a test-time adaptation method.

**Implementation details.** Aligned with previous work (Shu et al., 2022; Karmanov et al., 2024), we utilize publicly available pretrained CLIP models (Radford et al., 2021a) for image recognition, adopting ViT-B/16 (Dosovitskiy, 2020) backbones as the visual encoder and a Transformer (Vaswani, 2017) as the text encoder. We also employ the pretrained ULIP model (Xue et al., 2023) for point cloud recognition. For our C-TTA, we employ hand-crafted prompts following (Karmanov et al., 2024), setting $h = 20$ (Eq. 8) and $w = 0.01$ (Eq. 10). During testing, we strictly adhere to the TTA setting in (Shu et al., 2022), using a batch size of 1. We report top-1 accuracy as the evaluation metric. All experiments are conducted on an NVIDIA 3090 GPU.

## 4.2 COMPARISONS WITH STATE-OF-THE-ART

**Results on the Cross-Datasets Generalization.** We report the quantitative results of various methods for cross dataset generalization on 10 benchmarks. Table 1 shows that our C-TTA achieves the highest average accuracy, demonstrating strong competitiveness. In particular, our C-TTA surpasses

Table 2: Comparison of GS-Bias in Domain Generalization using ViT-B/16 Dosovitskiy (2020) as the backbone. Bold values indicate the best performance. The two evaluation metrics, Average and OOD Average, are computed by calculating the mean accuracy across all five datasets, as well as the four OOD datasets, excluding ImageNet.

| Method | ImageNet | ImageNet-A | ImageNet-V2 | ImageNet-R | ImageNet-S | Average | OOD Average |
|---|---|---|---|---|---|---|---|
| CLIP-ViT-B/16 | 66.73 | 47.87 | 60.86 | 73.98 | 46.09 | 57.20 | 59.11 |
| CoOp | **71.51** | 49.71 | 64.20 | 75.21 | 47.99 | 61.72 | 59.28 |
| CoCoOp | 71.02 | 50.63 | 64.07 | 76.18 | 48.75 | 62.13 | 59.91 |
| MaPLe | 70.72 | 50.90 | 64.07 | 76.98 | 49.15 | 62.36 | 60.69 |
| TPT | 68.98 | 54.77 | 63.45 | 77.06 | 47.94 | 62.44 | 60.81 |
| DiffTPT | 70.30 | 55.68 | 65.10 | 75.00 | 46.80 | 62.28 | 60.52 |
| ZERO | 69.06 | **61.35** | 64.13 | 77.28 | 48.29 | 64.02 | 62.76 |
| MTA | 70.08 | 58.06 | 64.24 | 78.33 | 49.61 | 64.06 | 62.56 |
| GS-Bias | 70.57 | 56.61 | 64.62 | 80.49 | 50.33 | 64.52 | 63.01 |
| TDA | 69.51 | 60.11 | 64.67 | 80.24 | 50.54 | 65.01 | 63.89 |
| BCA | 70.22 | 61.14 | **64.90** | 80.72 | 50.87 | 65.37 | 64.16 |
| C-TTA | 70.28 | 61.20 | 64.85 | **80.79** | **50.95** | **65.61** | **64.45** |

training-time adaptation methods that rely on labeled data, such as outperforming the state-of-the-art MaPLe (Khattak et al., 2023) by 3.15%, indicating that test-time adaptation offers a more practical solution. Compared to existing TTA approaches, our C-TTA consistently achieves state-of-the-art performance. For clarity, we categorize prior works into instance-wise TTA and episodic-wise TTA.

In instance-wise TTA, methods such as TPT (Shu et al., 2022) and DiffTPT (Feng et al., 2023) rely on entropy minimization to optimize text prompts. However, the narrow focus on this objective often results in overconfident predictions. More recent approaches, such as MTA (Zanella & Ben Ayed, 2024) and ZERO (Farina et al., 2024), shift toward training-free adaptation, thereby avoiding reliance on explicit optimization objectives. Nevertheless, these methods are fundamentally built on the assumption of a static target domain, which is inconsistent with the dynamic distributions encountered in real-world scenarios. In contrast, our C-TTA surpasses all of them across 10 benchmark datasets, achieving improvements of 4.83% over MTA and 4.36% over TPT, respectively.

Shifting to episodic-wise TTA, TDA (Karmanov et al., 2024) treats test data as a streaming input and updates predictions using a cache of selected high-confidence and high-uncertainty samples. While effective, this approach inherently restricts adaptation to only a fraction of the target domain. In contrast, our C-TTA exploits all incoming test samples to continuously accumulate target-domain knowledge, delivering an average improvement of 1.93% over TDA. Notably, C-TTA also surpasses DPE (Zhang et al., 2024a), despite DPE's heavy reliance on gradient-based optimization, cache mechanisms, and numerous augmented views. Distinctively, C-TTA is training-free, cache-free, and augmentation-free, striking a unique balance between performance and efficiency that makes it highly practical for real-world deployment.

**Results on the Domain Generalization.** We further assess the generalization capacity of C-TTA across domains on ImageNet and its four variants. As shown in Table 2, Our C-TTA consistently outperforms all instance-wise TTA approaches in both average accuracy and OOD performance, due to its ability to continuously model target-domain knowledge. For example, C-TTA achieves a 1.44% improvement over GS-Bias (Huang et al., 2025) in OOD scenarios, highlighting that leveraging latent relationships across test samples is more effective than relying on each sample in isolation. Furthermore, compared with episodic-wise approaches such as TDA (Karmanov et al., 2024), C-TTA demonstrates consistent gains across all datasets, underscoring the advantage of accumulating knowledge from all test samples rather than from a limited subset. It is also worth noting that training-time adaptation methods exhibit a natural advantage on ImageNet, which is closer to

Table 3: Comparison of recognition accuracy on ModelNet-C (Ren et al., 2022) and 3 corrupted variants of ScanObjectNN-C (Uy et al., 2019) that includes 7 types of corruptions. Results are reported for a corruption severity level of 2. Each clean point cloud contains 1024 points. The last column is the average across the 7 types of corruptions.

| Method | Corruption Type | | | | | | | Avg. |
|---|---|---|---|---|---|---|---|---|
| | Add Global | Add Local | Drop Global | Drop Local | Rotate | Scale | Jitter | |
| *ModelNet-C* | | | | | | | | |
| ULIP | 33.55 | 43.92 | 54.70 | 50.89 | 55.27 | 50.20 | 44.08 | 47.52 |
| Point-Cache | 46.15 | 47.85 | 59.16 | 56.00 | **61.47** | 55.35 | 48.91 | 53.70 |
| C-TTA | **48.78** | **51.17** | **61.54** | **56.69** | 61.42 | **56.44** | **49.88** | **55.13** |
| *ScanObjectNN (OBJ-ONLY)* | | | | | | | | |
| ULIP | 31.50 | 34.77 | 51.29 | 38.38 | 48.36 | 44.58 | 36.83 | 40.82 |
| Point-Cache | 32.01 | 38.04 | **54.56** | 45.27 | 50.95 | 45.96 | 39.24 | 43.72 |
| C-TTA | **34.76** | **39.76** | 53.53 | **45.78** | **51.81** | **46.47** | **39.59** | **44.53** |
| *ScanObjectNN (OBJ-BG)* | | | | | | | | |
| ULIP | 27.19 | 25.82 | 45.61 | 34.25 | 40.96 | 40.10 | 30.98 | 34.99 |
| Point-Cache | 28.23 | 30.12 | 48.71 | 40.45 | 43.55 | 40.28 | **34.42** | 37.97 |
| C-TTA | **32.36** | **30.98** | **52.15** | **42.00** | **48.53** | **46.64** | 32.19 | **40.69** |
| *ScanObjectNN (hardest)* | | | | | | | | |
| ULIP | 19.26 | 18.39 | 30.99 | 23.91 | 27.48 | 26.34 | 21.44 | 23.97 |
| Point-Cache | 23.46 | **22.69** | 34.70 | 31.75 | 33.00 | 28.28 | **25.05** | 28.42 |
| C-TTA | **24.18** | 22.07 | **38.41** | **32.51** | **34.21** | **30.78** | 23.63 | **29.40** |

the pretraining source domain, yet suffer significant performance degradation in OOD settings. In contrast, our test-time adaptation maintains a robust performance across both in-domain and OOD datasets.

**Results on Robust Point Cloud Analysis.** Beyond image classification, we further extend C-TTA to point cloud recognition to demonstrate its general applicability. As reported in Table 3, we evaluate its robustness on four corrupted point cloud datasets. Our method consistently improves the zero-shot performance of ULIP (Xue et al., 2023) across all datasets, with gains of +7.61% on ModelNet-C (Ren et al., 2022), +3.71% on ScanObjectNN (OBJ-ONLY), +5.70% on ScanObjectNN (OBJ-BG), and +5.43% on ScanObjectNN (hardest). Notably, C-TTA also outperforms Point-Cache (Sun et al., 2025), which enhances predictions by retrieving cached global and spatial point cloud features at test time. These results clearly indicate that leveraging continuous test samples for adaptation is more effective than relying on a partial subset of cache data, even in the context of 3D visual understanding.

**Efficiency and Effectiveness.** In this experiment, we adopt ViT as the visual backbone on a single NVIDIA 3090 GPU to evaluate the efficiency and effectiveness of the proposed C-TTA method on the ImageNet dataset. As shown in Figure 1(I)(c), our inference speed significantly outperforms both gradient-based and gradient-free methods. For example, TPT achieves only 1.1 FPS, and although TDA improves efficiency, it still lags behind the original CLIP. In contrast, our method requires no backpropagation or cache retrieval, achieving superior efficiency, running $5.7 \times$ faster than TDA. Moreover, as illustrated in Figure 2(a), C-TTA exhibits substantially lower memory usage compared to existing training-free approaches. This is because our method avoids additional forward passes for augmented views and reduces redundant matrix multiplications, resulting in memory consumption comparable to the original CLIP. These advantages in inference speed and memory efficiency make C-TTA highly practical for real-world deployment.

## 4.3 ABLATION STUDY

In this section, we provide a comprehensive analysis of the advantages of GS-Bias, exploring the impact of different hyperparameter settings. All ablation studies are conducted across 11 benchmark datasets, including ImageNet and 10 datasets in cross-dataset benchmark.

**The effects of $h$.** We study the effect of the temperature parameter $h$ in updating the history text features. As shown in Figure 2(b), performance first increases and then decreases as $h$ grows.

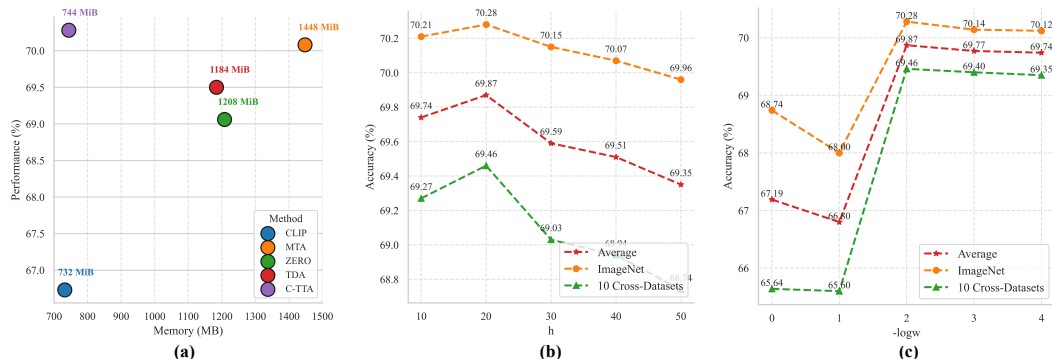

Figure 2: (a) Comparison of memory and performance between C-TTA and training-free methods on ImageNet. (b) Ablation study on the hyperparameter $h$ in Eq. 8. (c) Ablation study on the hyperparameter $w$ in Eq. 10.

Small $h$ leads to overly aggressive updates, causing instability, while large T results in insufficient adaptation, effectively ignoring current image features. The optimal $h$ balances historical knowledge and new information, yielding the best performance. This trend is consistent across datasets and highlights the importance of selecting an appropriate $h$ for stable and effective test-time adaptation.

**The effects of** $w$**.** We evaluate the effect of the interpolation factor $w$ in combining the original text features and the history text features. As shown in Figure 2(c), performance first increases and then slightly decreases as $w$ decreases from 1 to 0.0001. A large $w$ relies mostly on the original text features, limiting adaptation to current image features. A very small $w$ over-emphasizes historical features, potentially introducing accumulated noise. The optimal performance is achieved at an intermediate $w$, balancing the contribution of original and history features. This trend is consistent across multiple datasets.

## 5 LIMITATION

We further discuss the unexplored limitations of the proposed C-TTA, which will be the focus of our future work. Although C-TTA enables continuous target-domain knowledge modeling and achieves a favorable trade-off between performance and efficiency, one notable limitation is its reliance on empirically chosen hyperparameters. Extensive ablation studies across different datasets indicate that the optimal settings of C-TTA are largely consistent. However, these hyperparameters still need to be manually tuned for different test samples. While this empirical strategy provides a practical baseline, it may not guarantee optimal performance for every individual instance. Future work could explore dynamic parameter, such as the adaptive fusion weights discussed in Eq. 9 We believe that such extensions would further enhance the adaptability and generalization capability of the model.

## 6 CONCLUSION

In this work, we propose Continuous Test-Time Adaptation (C-TTA). Unlike previous approaches that focus only on instance-wise or episodic TTA, C-TTA leverages all available test samples, which is crucial for modeling continuously evolving real-world domain distributions. By adaptively collecting visual information from incoming samples through a simple target-prototype mechanism, C-TTA efficiently accumulates target-domain knowledge without additional training or heavy cache access. Our experiments demonstrate that C-TTA not only outperforms existing methods on image classification tasks but also generalizes effectively to point cloud analysis, making it a practical and robust solution for real-world scenarios.

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
