# OpenReview forum: "Continuous Test-Time Adaptation of Vision-Language Models"
_ICLR.cc/2026/Conference — ICLR 2026 Conference Withdrawn Submission_

### Official Review · Reviewer_UAvn · 2025-10-21

**Soundness:** 2
**Presentation:** 1
**Contribution:** 1
**Rating:** 2
**Confidence:** 5

**Summary:**

The paper argues that mainstream VLM TTA methods either assume a static target distribution (instance-wise prompt updates) or leverage only a small confident subset (episodic/cached), both failing to model continuous, evolving real-world distributions. It proposes C-TTA, a training-free, cache-free mechanism that continuously updates a class-confidence–weighted target prototype and combines this target distribution with the original CLIP distribution for prediction

**Strengths:**

- Simple, efficient mechanism: confidence-weighted EMA-style prototype update plus interpolation with text prototypes; no backprop, no cache.

**Weaknesses:**

- Positioning / VLM specificity. The core problem—failure to capture continuous, evolving target distributions—is not specifically for VLM; it is a modality-agnostic TTA/CL issue. This problem is not new, already discussed in our CL/TTA papers. Thus, the method (confidence-weighted prototype accumulation) also appears not only for VLM. The contribution would benefit from (i) sharper VLM-specific phenomena/analyses (e.g., text–image alignment drift, prompt-space dynamics), or (ii) an explicit modality-agnostic framing with supporting evidence beyond VLMs.

**Questions:**

- Which VLM-specific factors (if any) make C-TTA succeed where unimodal continuous TTA would fail? Any diagnostics on cross-modal alignment drift? (See weakness 1)

---

### Official Review · Reviewer_qxs2 · 2025-10-25

**Soundness:** 3
**Presentation:** 3
**Contribution:** 3
**Rating:** 4
**Confidence:** 5

**Summary:**

The authors propose a Continuous TTA (C-TTA) method to address adaptation of VLMs to test distributions in an online manner. C-TTA accumulates visual information from test samples, by updating prototypes, in a soft weighted manner. The final predictions is given out as a combination of probabilities from CLIP and the visual prototypes. The method is simple, effective from the results shown.

**Strengths:**

- The proposed method is simple and intuitive to understand. The paper is well presented and is easy to understand.
- The method is more efficient compared to prior methods like TPT, TDA.
- The experimental results show significant improvements compared to previous methods which are even more computationally expensive.
- The method is training-free and requires no cache memory to store features unlike prior methods.
- Extending beyond Image classification, to 3D point cloud data, expands the scope of this method.

**Weaknesses:**

- **Confusion in term C-TTA:** The term Continuous Test Time Adaptation is a very well established TTA setting where the test domains shift with time, for e.g., from rain->snow->fog, brightness->color-distortion->impulse-noise (CIFAR-10C/ImageNet-C corruptions) as studied in works [1,2]. However, here it not clear if this is indeed the setting followed. "Continuous real-world distributions" is mentioned several times, but the experimental details do not describe this original CTTA setting. The results are on individual datasets in Table 1. They are not indicative of domain-changing real world test environments.Continuous-TTA in this work only means that the continuous test data is leveraged to make the predictions (as done in prior works like TDA), but not where the test domains are continuously changing. I suggest the authors to modify the paper, to make this distinction clearly, not use the term "Continuous real-world distributions" as it is a misleading narrative.

- **Episodic-TTA:** The works TDA,DPE,etc. are also single image TTA methods, which leverage information from continuous test stream, which is the classical definition of TTA. There is no concept of episodes in the problem-setting considered here. So, what does episodic-TTA mean here. A better convention would be to call all methods (TPT,DiffTPT,...,TCA, TDA, DPE,..,BCA) as Single-image TTA. Although not necessary, if further needed, differentiate instance-wise/no-cache(TPT,DiffTPT,...,TCA) and cache-based methods(TDA, DPE,..,BCA) rather.

- **Sensistivity to hyperparameters:** As the authors also acknowledge in limitations, the method is fairly sensistive and need manual picking of hyperparameters, limiting its use in unknown test environments.

- **Experiments in original CTTA setting:** While the motivation is to adapt to continuously changing real-world distributions in the original CTTA setting[1,2], there are no experiments presented in this CTTA setting. This is a very natural and realistic TTA setting and is important to show its effectiveness when the domains change. Datasets like CIFAR-10C/100C, ImageNet-C, DomainNet can be used to evaluate on single-image continuously domain changing test streams, as the classes remain the same across different corruptions, as done in [1,2]

[1] "Continual test-time domain adaptation.", CVPR 2022.

[2] "Robust Mean Teacher for Continual and Gradual Test-Time Adaptation", CVPR 2023

**Questions:**

- Please clarify the term C-TTA, Episodic-TTA or use a different name, as mention in the weaknesses above.
- Have the authors tried other forms of weighting function $\beta_j$ to update the prototypes? It would be good to explain the intuition behind choosing equation of this form.
- The prototypes are updated in two steps: (1) Update class prototype using image feature with adaptive weight $\beta_j$. (2) interpolate text and image prototypes. Even after doing this interpolation, the scores of CLIP text-classifiers and these new prototypes are combined. But, the text prototype is already incorporated in Step (2). Does this not make the step where probabilities combined redundant? An ablation study on these aspects would justify the need for these steps, adding clarity to the paper.

---

### Official Review · Reviewer_ByLA · 2025-10-28

**Soundness:** 3
**Presentation:** 3
**Contribution:** 3
**Rating:** 2
**Confidence:** 4

**Summary:**

The article addresses the problem of continuous TTA, where the goal is to continuously adapt a vision-language model as test data are processed. Targeting classification tasks, the proposed approach (C-TTA) initialises a prototype per target class. Given a test sample, each prototype is updated using the confidence that the sample belongs to a specific class. The final classifier is then obtained by interpolating the cached prototypes updated at test-time and the CLIP original text prototypes. Performances show the advantage of this approach w.r.t. various TTA  and continuous TTA strategies.

**Strengths:**

1. The approach is simple yet sound and effective. Computationally, it requires only to estimate the confidence of CLIP-based zero-shot classifier on the test sample and using these estimates to update the cache. This overhead is minimal w.r.t. standard CLIP inference, both in terms of memory (i.e., only the prototypes need to be stored) and speed (few additional operations).

2. The model achieves good results, outperforming competitors on average on both standard benchmarks (e.g., Tab. 1 and Tab. 2) as well as a 3D application on point clouds (Tab. 3). These results shows the flexibility and effectiveness of the approach. Coupling the effectiveness with its simplicity, the approach might constitute a reference model for the community.

**Weaknesses:**

1. The title is misleading. While the approach is indeed performing the task in a continuous manner, the title might hint that the setting of continuous TTA is proposed with this work. In the introduction, related work and experiments, the setting per-se is denoted as episodic TTA (e.g., 46-49) and the name of continuous TTA associated with the method (e.g., 51). However, the title/name of the approach does not convey any aspect of the technical component (e.g., use of prototypes): it would be better to revise it to avoid confusion.

2. Being in continuous TTA implies that the order in which data arrives impacts the results. For instance, we may have a non i.i.d. distribution (e.g., data of a single class for the first N test samples): is the model robust to such a case? Moreover, how does the order of the data impact the results? These aspects are not thoroughly discussed/analyzed in the manuscript, while still being of importance for understanding the stability of the approach.

3. Linked to the previous point: the gap between C-TTA and the second-best approach is rather small on average (e.g., <0.3 in Tab. 2, 0.06 in Tab. 1). Thus, the dependence on random aspects (e.g., order of data) should be accounted for, as well as reporting the number of runs used to compute the results. This is important for reproducibility as well as reporting a fair ranking.

4.  The results are obtained using hand-crafted prompts (315-315), following TDA. Are all competitors using these prompts? And what is their impact? This again would help readers understand which aspects of the improvement are linked only to methodological design choices of C-TTA.

5. The rationale behind the weighting of Eq. (8) is not clear. Is the function related to previous works, or were alternative mechanisms considered but led to worse results? This could clarify potential challenges of computing the importance of each sample within the weighting mechanism.

6. A similar criticisms goes for how to compute the similarity score for updating the prototypes. According to Eq. (7) and lines 212-214, the scores are computed using standard zero-shot CLIP and the dynamic weights for the prototypes are derived accordingly. Why not using the "old" stored prototypes with the prediction described in Eq. (12)? This is counterintuitive as the latter leads to better downstream performance.

6. A similar criticism goes for how to compute the similarity score for updating the prototypes. According to Eq. (7) and lines 212-214, the scores are computed using standard zero-shot CLIP, and the dynamic weights for the prototypes are derived accordingly. Why not use the "old" stored prototypes with the prediction described in Eq. (12)? This is counterintuitive, as the latter leads to better downstream performance.

7. Related: Fig. 2.c report values of w in -log scale, showing that high w (e.g., =1) lead to very low results compared to much lower ones (e.g., ~0.1, or the used 0.01). These results are surprising as it seems that prototypes cannot be trusted when updating the cache (Eq. (10)). A similar behaviour is observed regarding the hyperparameter h: for the best value (h=20), $\beta$ is in the range 0-0.05, thus preserving the original cache values. The latter is meaningful given that it is driven by the number of seen samples and their estimated probability. However, an interesting analysis would have been to compare how much the final prototypes differ from the original CLIP embeddings and from the ideal prototypes (i.e., prototypes obtained from averaging the representations of the ground-truth elements of a class). My intuition is that, given the modality gap, the scores and the resulting $\beta$ should be quite low, thus vectors might be extremely close to the original CLIP embeddings. If we also account that prototype predictions are further averaged with CLIP ones (Eq. (12)), this may question how clean the prototypes are. As a consequence, if updates are noisy, why not update the prototypes after performing inference (e.g., Eq. (10) after Eq. (12))?

Minor:

9. In lines 470-471 it is stated that hyperparameters need to be manually tuned for each test sample. I guess this is not the case as for the statement and values in 315-316, but the sentence might be rephrased to avoid confusion and the impression that this is what has been conducted in the work.

References:
[a] Liang, Victor Weixin, et al. "Mind the gap: Understanding the modality gap in multi-modal contrastive representation learning." Advances in Neural Information Processing Systems 35 (2022): 17612-17625.

**Questions:**

Following on the weaknesses above:

1. Will the title be aligned to the contribution? if so, how?
2. Is the model robust to data ordering and non i.i.d. cases?
3. How many runs were performed for obtaining the results?
4. How do hand-craft prompts impact the results?
5. How has been the weighting function derived?
6. How do the obtained embeddings deviate from CLIP ones?
7. How would the order of update-prediction steps (point 7 above) and using prototypes for estimating scores (opint 6 above) affect the results?

---

### Official Review · Reviewer_aV5Q · 2025-11-01

**Soundness:** 2
**Presentation:** 2
**Contribution:** 2
**Rating:** 4
**Confidence:** 5

**Summary:**

This paper proposes Continuous Test-time Adaptation (C-TTA) for Vision-Language Models (VLMs), aiming to handle continuous distribution shifts that occur during streaming inference. The method requires no gradient updates or memory cache, leading to significantly improved computational efficiency, reportedly 5.7 to 71.7 times faster inference than prior methods such as TPT and TDA, while maintaining competitive or superior accuracy.

**Strengths:**

S1. **High efficiency with a strong performance trade-off.**
The paper reports up to 5.7 times faster inference speed compared to existing SOTA methods such as TPT and TDA, which makes it highly practical for real-world applications (e.g., VLMs on edge devices or online streaming scenarios). In particular, the EMA-based update that leverages the entire test stream has an extremely small memory footprint and performs only a single-vector update per step, allowing real-time operation.

S2. **Comprehensive and systematic evaluation.**
The authors conduct thorough experiments across a wide range of datasets and include comparisons with all major TTA baselines. Both the qualitative visualizations and ablation studies are clear and well-structured, demonstrating a strong level of reproducibility.

S3. **Strong empirical stability and simplicity.**
The proposed method involves only a small number of hyperparameters, each with clear intuitive meaning. Its simple EMA-based update shows robust convergence and drift resistance, while removing backpropagation helps mitigate issues such as catastrophic forgetting and overfitting to noisy samples.

**Weaknesses:**

W1. **Insufficient justification for hyperparameter stability.**
The paper fixes several hyperparameters throughout all experiments, but provides no sufficient sensitivity analysis to justify this choice.
In scenarios with large domain gaps or rapid distribution shifts, the optimal value of a fixed $\alpha$ is unlikely to remain consistent.
It would strengthen the paper if the authors discussed potential alternatives such as adaptive $\alpha$ scheduling or a decay mechanism to improve robustness across different test-time conditions.

W2. **Limited analysis of robustness to sample order and outliers.**
In the TTA-VLM literature, it is standard practice to report the mean and standard deviation over multiple runs to assess performance consistency. Given that the proposed continuous TTA method updates prototypes sequentially, its performance may vary depending on the sample order and inherent randomness in the data stream. Therefore, it would be important to repeat experiments multiple times to quantify this variance and to demonstrate the method’s robustness to sample ordering and outlier contamination.

W3. **Lack of specific results regarding efficiency.**
In Figure 1, the authors claim that their approach achieves approximately 5.7 times faster inference while maintaining higher accuracy compared to baselines. However, the main tables (Tables 1 and 2) only report accuracy results without providing per-dataset inference speed. For the sake of logical completeness and fair comparison, it would be helpful to include the inference time or throughput metrics directly in the main tables to substantiate the claimed efficiency advantage.

**Questions:**

Q1. **Lack of evaluation under long-tailed or class-imbalanced scenarios.**
Since the target prototype is updated only using a limited number of high-confidence samples, it may easily become biased toward head classes. The paper does not include comparisons with recent Long-tailed Test-time Adaptation (LTTA) methods, nor does it present per-class (head vs. tail) performance breakdowns. It would be valuable to demonstrate that the proposed continuous adaptation remains stable and effective under class imbalance or skewed label distributions.

Q2. **Justification for the surprising effectiveness of EMA smoothing.**
The main contribution of this paper appears to be the empirical finding that a simple EMA smoothing alone can achieve stable performance under continuous distribution shifts. However, such a simple mechanism leading to significant performance gains calls for a stronger theoretical or empirical justification. Could the authors clarify why EMA smoothing is particularly effective in the TTA setting for VLMs? For instance, which properties of the test-time distribution or representation space make such averaging especially impactful?
While the reviewer finds the proposed simple yet effective approach impressive, the fact that EMA smoothing alone yields substantial gains on continuous TTA benchmarks feels almost magical. A convincing justification for this phenomenon would significantly influence the reviewer’s final evaluation of this work.

---

### Note · Authors · 2025-11-21

I have read and agree with the venue's withdrawal policy on behalf of myself and my co-authors.